# In Silico Identification of Potential Natural Product Inhibitors of Human Proteases Key to SARS-CoV-2 Infection

**DOI:** 10.3390/molecules25173822

**Published:** 2020-08-22

**Authors:** R.P. Vivek-Ananth, Abhijit Rana, Nithin Rajan, Himansu S. Biswal, Areejit Samal

**Affiliations:** 1The Institute of Mathematical Sciences (IMSc), Chennai 600113, India; vivekananth@imsc.res.in (R.P.V.-A.); nithinofficial2912@gmail.com (N.R.); 2Homi Bhabha National Institute (HBNI), Mumbai 400094, India; abhijit.rana@niser.ac.in; 3School of Chemical Sciences, National Institute of Science Education and Research (NISER), Bhubaneswar 752050, India

**Keywords:** COVID-19, TMPRSS2, cathepsin L, molecular docking, molecular dynamics, non-covalent interactions, phytochemical inhibitors

## Abstract

Presently, there are no approved drugs or vaccines to treat COVID-19, which has spread to over 200 countries and at the time of writing was responsible for over 650,000 deaths worldwide. Recent studies have shown that two human proteases, TMPRSS2 and cathepsin L, play a key role in host cell entry of SARS-CoV-2. Importantly, inhibitors of these proteases were shown to block SARS-CoV-2 infection. Here, we perform virtual screening of 14,011 phytochemicals produced by Indian medicinal plants to identify natural product inhibitors of TMPRSS2 and cathepsin L. AutoDock Vina was used to perform molecular docking of phytochemicals against TMPRSS2 and cathepsin L. Potential phytochemical inhibitors were filtered by comparing their docked binding energies with those of known inhibitors of TMPRSS2 and cathepsin L. Further, the ligand binding site residues and non-covalent interactions between protein and ligand were used as an additional filter to identify phytochemical inhibitors that either bind to or form interactions with residues important for the specificity of the target proteases. This led to the identification of 96 inhibitors of TMPRSS2 and 9 inhibitors of cathepsin L among phytochemicals of Indian medicinal plants. Further, we have performed molecular dynamics (MD) simulations to analyze the stability of the protein-ligand complexes for the three top inhibitors of TMPRSS2 namely, qingdainone, edgeworoside C and adlumidine, and of cathepsin L namely, ararobinol, (+)-oxoturkiyenine and 3α,17α-cinchophylline. Interestingly, several herbal sources of identified phytochemical inhibitors have antiviral or anti-inflammatory use in traditional medicine. Further in vitro and in vivo testing is needed before clinical trials of the promising phytochemical inhibitors identified here.

## 1. Introduction

In December 2019, a new respiratory disease with unknown cause with clinical symptoms of fever, cough, shortness of breath, fatigue and pneumonia was first reported in Wuhan, (China) [1,2,3]. While most cases of this new disease show mild to moderate symptoms, a small fraction of cases, especially those with comorbid conditions like diabetes and hypertension, can develop fatal conditions such as acute respiratory distress syndrome (ARDS) due to severe lung damage [4]. In January 2020, a novel betacoronavirus, initially named 2019-nCoV, was discovered to be the etiological agent of this new disease [1,2,3]. Subsequently, human-to-human transmission of this disease was confirmed [4]. By 30 January 2020, the 2019-nCoV had spread to more than 20 countries and the World Health Organization (WHO) declared a public health emergency of international concern. On 11 February 2020, the international committee on taxonomy of viruses permanently named 2019-nCoV as severe acute respiratory syndrome coronavirus 2 (SARS-CoV-2) and the WHO named the associated disease as coronavirus disease 2019 (COVID-19). By 11 March 2020, COVID-19 had spread to more than 150 countries across six continents and the WHO upgraded the status of the epidemic to pandemic. As of 27 July 2020, the number of laboratory confirmed COVID-19 cases and deaths have already surpassed 16 million and 650,000, respectively, with the worst affected countries as USA, Brazil, UK, Mexico, Italy, India, France and Spain (https://www.worldometers.info/coronavirus/). In short, the COVID-19 pandemic is an unprecedented public health and economic emergency for humankind.

Coronaviruses are enveloped, positive-sense, single-stranded RNA viruses with large viral genomes. The publication of SARS-CoV-2 genome in January 2020 led to its taxonomic classification into the family *Coronaviridae* and genus *Betacoronavirus* [1,2,3]. Bats are natural reservoirs of coronaviruses [5]. The SARS-CoV-2 genome shares 96% nucleotide identity with bat coronavirus RaTG13, which suggests a probable zoonotic transfer to humans via an intermediate animal host [6]. Prior to SARS-CoV-2 epidemic, there are two precedence of zoonotic transfer of *betacoronaviruses* to humans, namely the severe acute respiratory syndrome coronavirus (SARS-CoV) and the middle east respiratory syndrome coronavirus (MERS-CoV), which had also led to outbreaks of severe respiratory disease [7]. In 2002–2003, SARS-CoV emerged in China and led to 8098 infections and 774 deaths across the world [7]. Interestingly, the SARS-CoV-2 genome shares ~80% nucleotide identity with SARS-CoV [1,2,3]. In 2012, MERS-CoV emerged in Saudi Arabia and led to 2521 infections and 866 deaths that were largely limited to Middle Eastern countries [7]. Unlike SARS and MERS, the geographic spread of COVID-19 and the ensuing mortality is significantly higher. To date, there are no approved antiviral drugs or vaccines against *betacoronavirus* infections including COVID-19 [8]. Hence, the current measures to contain COVID-19 include social distancing, aggressive testing, patient isolation, contact tracing and travel restrictions. In present circumstances, an immediate goal of biomedical research is to develop antivirals or anti-COVID therapeutics for SARS-CoV-2 [8].

The SARS-CoV-2 genome comprises ~30,000 bases with 14 open reading frames (ORFs) coding for 27 proteins [9]. The genome organization of SARS-CoV-2 is similar to other *betacoronaviruses* with the 5′-region coding for non-structural proteins and the 3′-region coding for structural proteins. Important structural proteins of SARS-CoV-2 coded by the 3′-region include the spike (S) surface glycoprotein, the envelope (E) protein, the matrix (M) protein and the nucleocapsid (N) protein [9]. The 5′-region of the SARS-CoV-2 genome contains the replicase gene which codes for two overlapping polyproteins, pp1a and pp1ab, which are proteolytically cleaved by two important non-structural proteins, 3-chymotrypsin like protease (3CL^pro^) and papain-like protease (PL^pro^), to produce functional (non-structural) proteins. Other important non-structural proteins of SARS-CoV-2 for the viral life cycle include the RNA-dependent RNA polymerase (RdRp) and helicase. Accordingly, the 4 non-structural proteins, 3CL^pro^, PL^pro^, RdRp and helicase, along with the spike glycoprotein of SARS-CoV-2 are among the most attractive targets for anti-COVID drugs [8].

Rather than targeting important SARS-CoV-2 proteins for viral life cycle, an alternative approach to anti-COVID drugs involves targeting host factors key to SARS-CoV-2 infection [8]. For host cell entry, SARS-CoV-2 employs the spike (S) protein whose S1 subunit has a receptor binding domain (RBD) that specifically recognizes the cell surface receptor angiotensin converting enzyme 2 (ACE2) [10,11,12,13,14]. Notably, both SARS-CoV-2 and SARS-CoV employ ACE2 as the cell entry receptor [10,11,12,13,14]. After attachment of S protein to ACE2 receptor, the membrane fusion of virus and host cell depends on proteolytic activation of S protein by host proteases which involves cleavage of S1 subunit at S1/S2 and S2′ sites resulting in dissociation of S1 subunit and structural change in S2 subunit of S protein [10,11,12,13,14]. Hoffmann et al. [10] showed that the host cell proteases, Transmembrane Protease Serine 2 (TMPRSS2) and cathepsin L or cathepsin B, can carry out S protein priming required for SARS-CoV-2 entry. Hoffmann et al. [10] also showed that TMPRSS2 is more essential for S protein priming and SARS-CoV-2 entry. In parallel, Ou et al. [11] used specific inhibitors of cathepsin L and cathepsin B to show that cathepsin L rather than cathepsin B is essential for S protein priming of SARS-CoV-2 and membrane fusion in lysosomes. These studies highlight at least two alternate pathways for host cell entry of SARS-CoV-2. On the one hand, after SARS-CoV-2 attachment to ACE2, the membrane fusion and cytoplasmic entry can occur at the plasma membrane provided the cell surface protease TMPRSS2 is available to carry out S protein priming. On the other hand, after SARS-CoV-2 attachment to ACE2, the virus can be internalized as part of endosomes in the endocytic pathway, and later, the membrane fusion and cytoplasmic entry will occur in lysosomes provided the lysosomal protease cathepsin L is available to carry out S protein priming [10,11,12]. Depending on the target cell and associated expression of host cell proteases, SARS-CoV-2 may use one of the alternative pathways for host cell entry. Importantly, the above-mentioned studies also showed that known inhibitors camostat mesylate and nafamostat mesylate of TMPRSS2 and E-64d and PC-0626568 (SID26681509) of cathepsin L can block or significantly reduce the host cell entry of SARS-CoV-2 [10,11,12]. In conclusion, human proteases TMPRSS2 and cathepsin L are key factors for host cell entry and are important targets for anti-COVID drugs [10,11,12,15].

To expedite this search for anti-COVID drugs, several computational studies have used homology modeling or published crystal structures of SARS-CoV-2 proteins, molecular docking and diverse small molecule libraries, to predict potential inhibitors of SARS-CoV-2 proteins including among existing approved drugs for repurposing and natural compounds (see e.g., [16,17,18,19,20,21,22,23,24]). In comparison, fewer computational studies [16,19] have focussed on identification of potential inhibitors of host factors. In this work, we perform virtual screening of a large phytochemical library specific to Indian medicinal plants to identify potential natural product inhibitors of TMPRSS2 and cathepsin L.

Plant-based natural products have made immense contributions to drug discovery [25]. Specifically, ~40% of the small-molecule drugs approved to date by the US Food and Drug Administration (FDA) are either natural products or natural product derivatives. Recently, there are reports from China of successful use of traditional Chinese medicine and associated herbs in treatment of COVID-19 patients [26]. On similar lines, there have been suggestions to tap the rich legacy of traditional Indian medicine and information on phytochemicals of Indian herbs in the search for anti-COVID drugs [27]. Previously, some of us have built IMPPAT, the largest resource to date on phytochemicals of Indian herbs [28]. In this work, we perform molecular docking using a large library of 14,011 phytochemicals compiled mainly from IMPPAT to identify potential natural product inhibitors of TMPRSS2 and cathepsin L. Subsequently, we have also performed molecular dynamics (MD) simulations to investigate the stability of protein-ligand complexes for the top phytochemical inhibitors of TMPRSS2 and cathepsin L identified in this work.

## 2. Results

### 2.1. Workflow for Virtual Screening

This computational study aims to predict potential phytochemical inhibitors of human proteases, TMPRSS2 and cathepsin L, that are important for priming of S protein and cell entry of SARS-CoV-2 [10,11,12]. Briefly, the workflow for this virtual screening is as follows (Figure 1).

In the first stage, we prepared the ligands for molecular docking with target proteases. We compiled a library of 14,011 phytochemicals produced by medicinal plants used in traditional Indian medicine, and the main source of this compilation was IMPPAT [28], the largest resource on phytochemicals of Indian herbs to date (see Section 3). Next, the standard drug-likeness measure, Lipinski’s rule of five (RO5) [29], was used to filter a subset of 10,510 drug-like phytochemicals. Next, the filtered phytochemicals were prepared for docking by retrieving their 3D structures from Pubchem [30] followed by energy-minimization using OpenBabel [31] (Section 3).

In the second stage, we prepared the target proteins for docking with prepared ligands. For TMPRSS2, the 3D structure is yet to be determined experimentally, and thus, we built a homology model of TMPRSS2 using SWISS-MODEL [32] which was used for docking after energy-minimization using UCSF Chimera [33] (Figure 2; Section 3). Figure 2 displays the TMPRSS2 model structure with the catalytic triad S441, H296 and D345 and the substrate binding residue D435 in the S1 subsite. For cathepsin L, the crystal structure (PDB 5MQY) with 1.13 Å resolution was used for docking after energy-minimization using UCSF Chimera (Figure 3; Section 3). Figure 3 displays the cathepsin L structure with the catalytic dyad C25 and H163 in S1 subsite, and other important residues in S2 and S1′ subsites.

In the third stage, we performed protein-ligand docking using AutoDock Vina [34]. For protein-ligand docking, an appropriate grid box was manually determined for TMPRSS2 and cathepsin L (Section 3). To decide on a stringent binding energy cut off for the identification of potential inhibitors, docking was first performed for known inhibitors of target proteins (Section 3). Based on the docking binding energies of the known inhibitors, camostat and nafamostat, to TMPRSS2, we decided on a stringent criteria of binding energy ≤−8.5 kcal/mol for the best docked pose of screened ligands to identify potential inhibitors of TMPRSS2 (Section 3). Similarly, based on the docking binding energies of the known inhibitors, E-64d and PC-0626568, to cathepsin L, we decided on a stringent criteria of binding energy ≤−8.0 kcal/mol for the best docked pose of screened ligands to identify potential inhibitors of cathepsin L (Section 3). Thereafter, docking was performed for the prepared ligands in the phytochemical library against the prepared structures of TMPRSS2 and cathepsin L (Section 3). Lastly, we filtered the subset of phytochemicals whose binding energy in the best docked pose with TMPRSS2 (respectively, cathepsin L) is ≤−8.5 kcal/mol (respectively, ≤−8.0 kcal/mol). Moreover, the best docked pose with TMPRSS2 or cathepsin L of each filtered phytochemical was separated from AutoDock Vina output file, and then, combined with the target protein structure to obtain the docked protein-ligand complex in .pdb format (Section 3). At the end of third stage, we obtained 101 phytochemicals whose binding energy in the best docked pose with TMPRSS2 is ≤−8.5 kcal/mol and 16 phytochemicals whose binding energy in the best docked pose with cathepsin L is ≤−8.0 kcal/mol.

In the fourth stage, the structure of docked protein-ligand complex in .pdb format for each filtered phytochemical from third stage was used to determine the ligand binding site residues in the target protein and different non-covalent interactions such as hydrogen bond, halogen bond, hydrophobic interactions, etc. between ligand and target protein (Section 3). In case of TMPRSS2, the specificity of this trypsin-like protease is determined by the conserved substrate binding residue D435 in the S1 pocket [35] (Section 3), and therefore, a potent inhibitor should either bind to or form non-covalent interactions with D435. In case of cathepsin L, the specificity of this cysteine protease is dependent on the catalytic residues, C25 and H163, in the S1 subsite (Section 3), and therefore, a potent inhibitor should either bind to or form non-covalent interactions with the catalytic residues. In this work, we consider a phytochemical to be a potential inhibitor of TMPRSS2 (respectively, cathepsin L) only if the ligand binding energy in the best docked pose is ≤−8.5 kcal/mol (respectively, ≤−8.0 kcal/mol) and the ligand binds to or forms non-covalent interactions with the residue D435 in TMPRSS2 (respectively, residues C25 and H163 in cathepsin L).

At the end of fourth stage, we obtained 96 phytochemicals (labelled T1–T96; Figure 5; Appendix A) as potential inhibitors of TMPRSS2 and 9 phytochemicals (labelled C1–C9; Figure 6; Appendix A) as potential inhibitors of cathepsin L. Using IMPPAT [28], we provide a list of Indian medicinal plants that can produce the identified phytochemical inhibitors of TMPRSS2 and cathepsin L (Table 1 and Table 2; Appendix A). Furthermore, we have also compiled information on potential antiviral or anti-inflammatory use in traditional medicine of the herbal sources of the identified phytochemical inhibitors of TMPRSS2 and cathepsin L (Table 1 and Table 2; Appendix A). In Appendix A, we list the ligand binding site residues and non-covalent protein-ligand interactions for the identified phytochemical inhibitors of TMPRSS2 and cathepsin L. We have also predicted the physicochemical and ADMET properties of the identified phytochemical inhibitors of TMPRSS2 and cathepsin L (Section 3; Appendix A).

Finally, in the fifth stage, for the top three inhibitors of TMPRSS2 namely, T1 (qingdainone), T2 (edgeworoside C) and T3 (adlumidine), and of cathepsin L namely, C1 (ararobinol), C2 ((+)-oxoturkiyenine) and C3 (3α,17α-cinchophylline), their respective protein-ligand complexes were analyzed using 180 ns MD simulation (Section 3; Figures 8 and 9; Appendix A) and their interaction binding energy was computed using MM-PBSA method (Section 3; Table 3).

### 2.2. Potential Phytochemical Inhibitors of TMPRSS2

As mentioned above, we have identified 96 potential natural product inhibitors of TMPRSS2 by computational screening of 14,011 phytochemicals produced by Indian medicinal plants, and these 96 compounds labelled T1-T96 are listed in Appendix A along with their PubChem identifier, common name, IUPAC name and structure in SMILES format. In this section, we provide a detailed discussion of the top nine phytochemical inhibitors (labelled as T1–T9) whose binding energies in the best docked poses with TMPRSS2 are ≤−9.2 kcal/mol. Figure 4 displays the structure of these top 9 phytochemical inhibitors of TMPRSS2 and Table 1 provides a list of Indian medicinal plants that can produce them.

Phytochemical T1, qingdainone, has a binding energy of −9.6 kcal/mol. T1 is a quinazoline alkaloid produced by *Strobilanthes cusia*, a herb with antiviral activity [36]. Figure 5a shows the TMPRSS2 residues that form hydrogen bonds or π-π stacking interactions with T1. TMPRSS2 residue D440 forms C-H⋯O type hydrogen bond with T1 whereas residue A399 forms C-H⋯N type hydrogen bond with T1. Further, T1 forms hydrophobic contacts with residues I381, S382, T387, E388, N398, A400, D440, C465 and A466.

Phytochemical T2, edgeworoside C, also has a binding energy of −9.6 kcal/mol. T2 is a coumarin produced by *Edgeworthia gardneri*, a medicinal plant consumed as a herbal tea in Tibet [37]. In traditional medicine, *Edgeworthia gardneri* has been used to treat metabolic disorders including diabetes [38,39]. Figure 5b shows the TMPRSS2 residues that form hydrogen bonds or π-π stacking interactions with T2. T2 forms 12 hydrogen bonds with residues A386, N398, A399, V434, D435 and D440 of TMPRSS2. The phenolic hydroxyl group of T2 acts as both acceptor and donor forming O-H⋯O and N-H⋯O type hydrogen bonds with the substrate binding residue D435 and C-H⋯O type hydrogen bond with residue V434. The hydroxyl groups attached to the pyran ring of T2 form hydrogen bonds with residues A386, N398 and D440. Further, T2 forms hydrophobic contacts with residues E260, I381, A400, N433, and A466.

Phytochemical T3, adlumidine, also has a binding energy of −9.6 kcal/mol. T3 is produced by *Fumaria indica*, a herb used in traditional medicine to treat fever, cough, skin ailments and urinary diseases [40]. Adlumidine has also been reported to be an inhibitor of GABA receptor [41]. Figure 5c shows the TMPRSS2 residues that form hydrogen bonds or π-π stacking interactions with T3. The two 1,3-dioxole groups present in T3 facilitate the formation of an extensive hydrogen bond network with E388, E389, S436, C465 and A466. T3 also forms C-H⋯S type hydrogen bond [42] with C437. Further, T3 forms hydrophobic contacts with residues E260, I381, S382, T387, N398, A399 and A400.

Phytochemicals T4 (pseudo-α-colubrine), T6 (strychnine *N*-oxide), T7 (α-colubrine) and T9 (2-hydroxy-3-methoxystrychnine) have binding energies of −9.3 kcal/mol, −9.3 kcal/mol, −9.2 kcal/mol and −9.2 kcal/mol, respectively. These four phytochemicals are monoterpenoid indole alkaloids produced by *Strychnos nux-vomica*. The herb *Strychnos nux-vomica* is used in traditional Indian medicine and its alkaloids have been shown to exhibit anti-inflammatory, anti-oxidant, antitumor and hepatoprotective activities [43]. Note that *Strychnos nux-vomica* is a poisonous plant whose seeds are extensively used in Ayurveda only after proper detoxification procedure called *Shodhana* described in Ayurvedic texts [43]. Figure 5d shows the TMPRSS2 residues that form hydrogen bonds or π-π stacking interactions with T4. T4 forms C-H⋯O type hydrogen bonds with residues A400, N433, D435 (substrate binding residue), C437 and D440. Further, T4 forms hydrophobic contacts with residues E260, I381, S382, T387, N398, A399, V434, D440 and A466. Figure 5f shows the TMPRSS2 residues that form hydrogen bonds or π-π stacking interactions with T6. T6 forms a C-H⋯N type hydrogen bond with residue N398. The substrate binding residue D435 also forms a C-H⋯O type hydrogen bond with T6. Further, T6 forms hydrophobic contacts with residues N398, A400, V434 and A466. Appendix A shows the TMPRSS2 residues that form hydrogen bonds or π-π stacking interactions with T7. T7 forms five C-H⋯O type hydrogen bonds with residues N433, D435 (substrate binding residue), C437 and D440. Further, T7 forms hydrophobic contacts with residues E260, T387, N398, A399, A400, V434 and A466. Appendix A shows the TMPRSS2 residues that form hydrogen bonds or π-π stacking interactions with T9. The phenolic hydroxyl group of T9 forms hydrogen bonds with residues S382 and A399. The substrate binding site D435 also forms a C-H⋯O type hydrogen bond with T9. Further, T9 forms hydrophobic contacts with residues E260, N398, A399, A400 and V434.

Phytochemical T5, bicuculline, has a binding energy of −9.3 kcal/mol, and it is a stereoisomer of T3. T5 is an isoquinoline alkaloid and is produced by herbs *Corydalis govaniana*, *Nerium oleander* and *Fumaria indica*. Bicuculline has also been reported to be a GABA receptor inhibitor [44]. Figure 5e shows the TMPRSS2 residues that form hydrogen bonds or π-π stacking interactions with T5. The two 1,3-dioxole groups present in T5 facilitate the formation of an extensive hydrogen bond network with residues E388, E389 and A400. The Furan-2-one ring also forms a C-H⋯O type hydrogen bond with C437. Further, T5 forms hydrophobic contacts with residues E260, T387, E388, N398, A399 and A466.

Phytochemical T8, egenine, has a binding energy of −9.2 kcal/mol. T8 is an isoquinoline alkaloid produced by *Fumaria vaillantii*. In traditional medicine, *Fumaria vaillantii* has been reported to exhibit antifungal, anti-inflammatory and anti-psychotic activities [45]. Appendix A shows the TMPRSS2 residues that form hydrogen bonds or π-π stacking interactions with T8. The two 1,3-dioxole groups present in T8 form hydrogen bonds with G383, E388, E389 and A400. One of the hydroxyl groups present in T8 forms C-H⋯O type hydrogen bond with residue C437. Further, T8 forms hydrophobic contacts with residues T387, A399, E388, N398, E260, and A466.

### 2.3. Potential Phytochemical Inhibitors of Cathepsin L

As mentioned above, we have identified nine potential natural product inhibitors of cathepsin L by computational screening of 14,011 phytochemicals produced by Indian medicinal plants, and these compounds labelled C1–C9 are listed in Appendix A along with their PubChem identifier, common name, IUPAC name and structure in SMILES format. Figure 6 displays the structure of these top nine phytochemical inhibitors of cathepsin L and Table 2 provides a list of Indian medicinal plants that produce them.

Phytochemical C1, ararobinol, has a binding energy of −8.9 kcal/mol. C1 is a bianthraquinone produced by herb *Senna occidentalis* used in Ayurveda. In traditional medicine, *Senna occidentalis* has been reported for antibacterial, antifungal, anti-inflammatory, anti-diabetic and anti-cancer activities [46]. Interestingly, there is a Chinese patent application [47] on potential use of ararobinol to treat human influenza virus infections, however, this suggests only a potential antiviral activity of C1 not specific to SARS-CoV-2 which further needs to be verified through in vitro and in vivo experimental studies. Figure 7a shows the cathepsin L residues that form hydrogen bonds or π-π stacking interactions with C1. Residues Q19 and A138 form C-H⋯N and C-H⋯O type of hydrogen bonds, respectively, with C1. Also, the residue W189 forms both face-to-edge and face-to-face type of π-π stacking interaction with C1. Further, C1 forms hydrophobic contacts with residues C25, G139, L144, H163 and W189.

Phytochemical C2, (+)-oxoturkiyenine, has a binding energy of −8.3 kcal/mol. C2 is an isoquinoline-derived alkaloid produced by the herb *Hypecoum pendulum* [48]. Figure 7b shows the cathepsin L residues that form hydrogen bonds or π-π stacking interactions with C2. The 2,5-dihydro-furan ring present in C2 forms two N-H⋯O type hydrogen bonds with residue Q19 and W189. The catalytic residue H163 forms N-H⋯O type hydrogen bond with C2. Also, the residue W189 forms a face-to-edge type of π-π stacking interaction with C2. Further, C2 forms hydrophobic contacts with residues G139, H140, H163 and W189.

Phytochemical C3, 3α,17α-cinchophylline, has a binding energy of −8.3 kcal/mol. C3 is a cinchophylline-type of alkaloid produced by the herb *Cinchona calisaya*. The *Cinchona* alkaloids have been reported for their antimicrobial, antiparasitic and anti-inflammatory activities [49]. Figure 7c shows the cathepsin L residues that form hydrogen bonds or π-π stacking interactions with C3. C3 forms 8 hydrogen bonds with residues of cathepsin L. One of the catalytic residue C25 forms C-H⋯S and N-H⋯S type hydrogen bonds with C3. The other catalytic residue H163 forms C-H⋯N and N-H⋯N type hydrogen bonds with C3. Further, one of the two pyrrole rings present in C3 forms hydrogen bond with residue G23. Lastly, M70 forms a C-H⋯S type hydrogen bond with C3. Further, C3 forms hydrophobic contacts with residues Q21, C22, L69, M70, A135 and W189.

Phytochemical C4, rugosanine B, has a binding energy of −8.2 kcal/mol. C4 is a cyclopeptide alkaloid produced by the bark of *Ziziphus rugosa* [50]. Various parts of *Ziziphus rugosa* have been reported for their antibacterial, antifungal, anti-inflammatory and analgesic activities [51]. Figure 7d shows the cathepsin L residues that form hydrogen bonds or π-π stacking interactions with C4. The pyrrole ring present in C4 forms a N-H⋯O type hydrogen bond with residue D162. Moreover, C4 forms C-H⋯O type hydrogen bonds with A138, D162 and L69. Also, the residue W189 forms a face-to-edge type of π-π stacking interaction with C4. Further, C4 forms hydrophobic contacts with residues G23, C25, G67, M70, A135, A138, D162, H163, G164, W189 and A214.

Phytochemical C5, trichotomine, has a binding energy of −8.2 kcal/mol. C5 is a bisindole alkaloid present in *Clerodendrum trichotomum*. *Clerodendrum trichotomum* has been reported for its use to treat cold, hypertension, rheumatism, dysentery and other inflammatory diseases [52]. Figure 7e shows the cathepsin L residues that form hydrogen bonds or π-π stacking interactions with C5. The carboxylic acid group present in C5 forms hydrogen bonds with residues Q19 and H163. The indole ring of C5 forms a N-H⋯N type hydrogen bond with residue G68. Further, C5 forms hydrophobic contacts with residues G23, C25, G67, G68, L69 and Y72.

Phytochemical C6, tectol, has binding energy of −8.1 kcal/mol. C6 is a naphthoquinone derivative [53] present in *Tectona grandis* and *Tecomella undulata*. *Tectona grandis* has been reported to have anti-inflammatory and antiparasitic activities [45]. *Tecomella undulata* has been used to treat syphilis and also reported to have anti-inflammatory and anti-HIV activities [54]. Additionally, Tectol has been reported to inhibit farnesyltransferase enzyme [55]. Figure 7f shows the cathepsin L residues that form hydrogen bonds or π-π stacking interactions with C6. The pyran group of C6 is involved in a C-H⋯O type hydrogen bond with residue L144. The catalytic residue C25 forms a C-H⋯S type hydrogen bond with C6. The other catalytic residue H163 forms a N-H⋯O type hydrogen bond with C6. Also, the residue W189 forms both face-to-face and face-to-edge type of π-π stacking interaction with C6. Further, C6 forms hydrophobic contacts with G23, L144 and W189.

Phytochemical C7, silymonin, has a binding energy of −8.1 kcal/mol. C7 is a flavanolignan [56] present in *Silybum marianum*. *Silybum marianum* has been used as a hepatoprotective agent and is reported to have anti-oxidant and anti-inflammatory activities [57]. Appendix A shows the cathepsin L residues that form hydrogen bonds or π-π stacking interactions with C7. C7 has four hydroxyl groups which help in the formation of an extensive network of hydrogen bonds with residues Q21, A138 and G139. C7 also forms two N-H⋯O type hydrogen bonds with H163 and W189. Also, the residue W189 forms a face-to-edge type of π-π stacking interaction with C7. Further, C7 forms hydrophobic contacts with residues G23, A138, L144, H163 and W189.

Phytochemical C8, picrasidine M, has a binding energy of −8.0 kcal/mol. C8 is a β-carboline alkaloid present in *Picrasma quassioides*. *Picrasma quassioides* has been reported to have antiviral and antifungal activities [28]. Appendix A shows the cathepsin L residues that form hydrogen bonds or π-π stacking interactions with C8. The carboxylic group of residue D162 forms two C-H⋯O type hydrogen bonds with C8. Also, residues M70 and G23 form hydrogen bonds of type C-H⋯S and C-H⋯O, respectively, with C8. Also, the residue W189 forms a face-to-edge type of π-π stacking interaction with C8. Further, C8 forms hydrophobic contacts with residues G23, L69, D162 and W189.

Phytochemical C9, trisjuglone, has a binding energy of −8.0 kcal/mol. C9 is a naphthoquinone present in *Juglans regia* (i.e., walnut). In traditional medicine, *Juglans regia* has been reported to have anti-inflammatory, antifungal and antimicrobial activities [45]. Appendix A shows the cathepsin L residues that form hydrogen bonds or π-π stacking interactions with C9. The benzoquinone moiety present in C9 forms two C-H⋯O type hydrogen bonds with residue Q21 and one N-H⋯O type hydrogen bond with W189. In contrast, the other catalytic residue H163 forms a N-H⋯O type hydrogen bond with C9. Also, the residue W189 forms a face-to-edge type of π-π stacking interaction with C9. Further, C9 forms hydrophobic contacts with residues Q21, G23, C25, L144 and W189.

### 2.4. Molecular Dynamics Simulation of Top Inhibitors

In order to investigate the stability of the protein-ligand complexes of the identified inhibitors, MD simulation of 180 ns was performed for top three inhibitors of TMPRSS2 namely, qingdainone (T1), edgeworoside C (T2) and adlumidine (T3), and of cathepsin L namely, ararobinol (C1), (+)-oxoturkiyenine (C2) and 3α,17α-cinchophylline (C3). Specifically, we have performed six 180 ns MD runs for protein-ligand complexes (TMPRSS2-T1, TMPRSS2-T2, TMPRSS2-T3, cathepsin L-C1, cathepsin L-C2 and cathepsin L-C3) and two 180 ns MD runs for free TMPRSS2 and cathepsin L proteins (Section 3). To access the stability of the six protein-ligand complexes, we have computed radius of gyration (R_g_) of the protein, root mean square deviation (RMSD) of the Cα atoms of the protein, root mean square fluctuations (RMSF) of the Cα atoms of the protein, RMSD of the ligand and finally distance of the center of mass of the ligand from the center of mass of the catalytic residues or substrate binding residues of the protein in complex with the ligand (Figure 8 and Figure 9).

The R_g_ value of TMPRSS2 in complex with T1, T2 and T3 remains largely stable throughout the MD simulation (Figure 8a). This implies that the top inhibitors of TMPRSS2 namely, T1, T2 and T3 do not induce any major structural changes to TMPRSS2 and TMPRSS2 remains structurally stable in complex with these inhibitors. TMPRSS2 in complex with T1, T2 and T3 has an average R_g_ value of 2.110 ± 0.022 nm, 2.096 ± 0.024 nm and 2.091 ± 0.021 nm, respectively. Further, RMSD value of the Cα atoms of TMPRSS2 in complex with T1, T2 and T3 become stable after 80 ns (Figure 8b). Over the 80 ns to 180 ns time interval, TMPRSS2 in complex with T1, T2 and T3 has an average RMSD value of 0.525 ± 0.023 nm, 0.501 ± 0.016 nm and 0.634 ± 0.018 nm, respectively. Lastly, Figure 8c shows the RMSF value per residue for TMPRSS2 in complex with T1, T2 and T3. R_g_, RMSD and RMSF values of TMPRSS2 in complex with T1, T2 and T3 closely follow R_g_, RMSD and RMSF values of TMPRSS2 free protein (Figure 8a–c; Appendix A). Appendix A show the superimposed snapshots at 120 ns, 140 ns and 160 ns of TMPRSS2-T1, TMPRSS2-T2 and TMPRSS2-T3 complexes, respectively. To further quantify the stability of the inhibitors T1, T2 and T3 bound to TMPRSS2, we have computed the RMSD of T1, T2 and T3 (Figure 8d) and distance of the center of mass of T1, T2 and T3 from the center of mass of the substrate binding residue D435 in TMPRSS2 (Figure 8e). Both RMSD of T1, T2 and T3 bound with TMPRSS2 and distance of the center of mass of T1, T2 and T3 from the center of mass of the substrate binding residue D435 becomes largely stable after 100 ns of the MD simulation (Figure 8d,e).

Also, R_g_ value of cathepsin L in complex with C1, C2 and C3 is stable throughout the MD simulation implying C1, C2 and C3 do not induce any major structural changes to cathepsin L and cathepsin L remains structurally stable in complex with these inhibitors (Figure 9a). Cathepsin L in complex with C1, C2 and C3 has an average R_g_ value of 1.700 ± 0.010 nm, 1.706 ± 0.007 nm and 1.705 ± 0.008 nm, respectively. Similarly, RMSD value of the Cα atoms of cathepsin L in complex with C1, C2 and C3 become largely stable after 80 ns (Figure 9b). Over the 80 ns to 180 ns time interval, cathepsin L in complex with C1, C2 and C3 has an average RMSD value of 0.281 ± 0.028 nm, 0.276 ± 0.022 nm and 0.270 ± 0.014 nm, respectively. Figure 9c shows the RMSF value per residue for cathepsin L in complex with C1, C2 and C3. As in the case of TMPRSS2, R_g_, RMSD and RMSF values of cathepsin L in complex with C1, C2 and C3 closely follow R_g_, RMSD and RMSF values of cathepsin L free protein (Figure 9a–c; Appendix A). Appendix A show the superimposed snapshots at 120 ns, 140 ns and 160 ns of cathepsin L-C1, cathepsin L-C2 and cathepsin L-C3 complexes, respectively. In order to quantify the stability of the inhibitors C1, C2 and C3 bound to cathepsin L we have also computed the RMSD of C1, C2 and C3 (Figure 9d) and distance of the center of mass of C1, C2 and C3 from the center of mass of the catalytic residues C25 (Figure 9e) and H163 (Figure 9f) in cathepsin L.

C1 has a largely stable RMSD after 120 ns of the MD simulation, C2 has the lowest and most stable RMSD in comparison with C1 and C3, and C3 shows a largely stable RMSD from 50 ns to 130 ns and from 150 ns to 170 ns of the MD simulation (Figure 9d). Distance of the center of mass of C1, C2 and C3 from the center of mass of the catalytic residues C25 and H163 in cathepsin L also remains largely consistent after 120 ns of the MD simulation (Figure 9e,f).

### 2.5. MM-PBSA Binding Energy of Top Inhibitors

Molecular Mechanics Poisson–Boltzmann Surface Area (MM-PBSA) is a widely used method to compute the binding energy of small molecules with biological macromolecules such as proteins [58]. Notably, the protein-ligand binding energy obtained using MM-PBSA method has been found to be more accurate than that obtained from protein-ligand docking [58]. Therefore, we have computed the binding energy for the top three inhibitors of TMPRSS2 and cathepsin L using MM-PBSA method. From the 180 ns MD simulation of the six protein-ligand complexes (TMPRSS2-T1, TMPRSS2-T2, TMPRSS2-T3, cathepsin L-C1, cathepsin L-C2 and cathepsin L-C3), 80 snapshots were obtained between 100 ns to 180 ns of the simulation at an interval of 1 ns along the trajectory, and thereafter, the 80 snapshots were used to compute the binding energy using g_mmpbsa (Section 3) [59,60]. The final binding energy is the sum of contributions from van der Waals, electrostatic, polar solvation, and solvent accessible surface area (SASA) energy components. The contribution of each of the above components to the binding energy of the top inhibitors is shown in Table 3.

Although T1, T2 and T3 have the same docked binding energy value of −9.6 kcal/mol to TMPRSS2, there are differences in their binding energy computed using MM-PBSA method. While TMPRSS2-T1 complex has the lowest binding energy value of −39.15 ± 2.799 kcal/mol, TMPRSS2-T2 and TMPRSS2-T3 complexes have binding energy value of −30.284 ± 3.585 kcal/mol and −27.386 ± 2.077 kcal/mol, respectively (Table 3). In case of cathepsin L, cathepsin L-C1, cathepsin L-C2 and cathepsin L-C3 complexes have binding energy value of −22.384 ± 3.420 kcal/mol, −20.577 ± 3.600 kcal/mol and −26.156 ± 3.433 kcal/mol, respectively (Table 3). Moreover, in comparison to binding energy obtained from docking using AutoDock Vina, binding energy for the TMPRSS2 and cathepsin L complexes with their top inhibitors computed using MM-PBSA method were found to be 2-fold to 4-fold lower, signifying even stronger binding (Table 1, Table 2 and Table 3)

## 3. Materials and Methods

### 3.1. Phytochemical Library and Drug-Likeness Evaluation

Previously, some of us have built the Indian Medicinal Plants, Phytochemistry And Therapeutics (IMPPAT) database [28] which is the largest resource on phytochemicals of Indian herbs to date. For this study, we compiled a ligand library of 14,011 phytochemicals by augmenting the information in IMPPAT [28] with additional information compiled from other literature sources [61,62,63,64,65,66,67,68,69,70]. Thereafter, the widely used drug-likeness measure, Lipinski’s rule of five (RO5) [29], was employed to filter the potential drug-like molecules within the ligand library of 14,011 phytochemicals. Specifically, 10,510 phytochemicals passed the R05 drug-likeness filter. We then retrieved the three-dimensional (3D) structures of these phytochemicals from Pubchem [30]. Next the 3D structures of the drug-like phytochemicals were energy-minimized using *obminimize* within the OpenBabel toolbox [31]. Finally, the energy-minimized 3D structures of ligands in .sdf format were converted to .pdb format using OpenBabel.

### 3.2. Homology Modeling of TMPRSS2 Structure

TMPRSS2 is a trypsin-like serine protease whose catalytic site consists of the triad Ser441 (S441), His296 (H296) and Asp345 (D345) [35]. It is well established that trypsin-like serine proteases cleave peptide bonds following positively charged amino acid residues such as arginine or lysine, and this specificity of the enzyme is determined by a negatively charged aspartate residue located at the bottom of its S1 pocket [71]. In TMPRSS2, this specificity is determined by the conserved negatively charged residue Asp435 (D435) at the bottom of the S1 pocket [35].

To date the 3D structure of TMPRSS2 has not been experimentally determined, and thus, we have used SWISS-MODEL [32] webserver (https://swissmodel.expasy.org/interactive) to perform homology modeling of TMPRSS2. We submitted the TMPRSS2 protein sequence (NCBI reference sequence NP_005647.3) to SWISS-MODEL and selected the crystal structure of human protein hepsin (PDB 1Z8G) [72] as the template to build the model structure (Figure 2a). Note that hepsin (PDB 1Z8G) is also a Type II transmembrane trypsin-like serine protease, and it shares 38% sequence similarity with TMPRSS2 (NP_005647.3) (Figure 2b). Subsequently, UCSF Chimera [33] was used to minimize the energy of the TMPRSS2 model structure obtained from SWISS-MODEL. Thereafter, the energy-minimized TMPRSS2 model structure was assessed using the structure assessment tool within SWISS-MODEL. In the TMPRSS2 model, 94.19% of the amino acid residues were found to be in the Ramachandran favoured regions in the Ramachandran plot (Figure 2c) and the model structure has a MolProbity [73] score of 1.50.

### 3.3. Protein Structure of Cathepsin L

We use the crystal structure (PDB 5MQY) [74] of human cathepsin L with 1.13Å resolution obtained from Protein Data Bank for virtual screening. UCSF Chimera was used to minimize the energy of the cathepsin L structure. Figure 3 displays the cathepsin L structure with important residues in S1, S2 and S1′ subsites of the enzyme [75]. Previous research has also revealed that S1 and S2 subsites of cathepsin L are important for the specificity of the enzyme [75,76]. In cathepsin L, the catalytic site consists of Cys25 (C25) and His163 (H163) in the S1 subsite, and Trp189 (W189) is at the center of the S1′ subsite [75] (Figure 3). In cathepsin L, the S2 subsite with important residues Asp162 (D162), Met161 (M161), Ala135 (A135), Met70 (M70) and Leu69 (L69) forms a deep hydrophobic pocket, and lastly, the conserved residue Gly68 (G68) is at the center of the S3 subsite [75,77] (Figure 3).

### 3.4. Molecular Docking

AutoDock Vina [34] was used to perform the molecular docking of energy-minimized 3D structures of ligands with energy-minimized structure of target proteins. Accordingly, the 3D structures of prepared ligands in .pdb format were converted to .pdbqt format using the python script *prepare_ligand4.py* from AutoDockTools [78]. Similarly, the energy-minimized structure of TMPRSS2 and cathepsin L in .pdb format were converted to .pdbqt format using the python script *prepare_receptor4.py* from AutoDockTools [78].

For protein-ligand docking, the appropriate grid box specified by the search space centre and search space size for TMPRSS2 and cathepsin L was manually determined by considering the key residues in target proteins such as the catalytic residues and substrate binding residues, which are important for the function and specificity of the considered proteases as reported in the literature. For TMPRSS2, the grid box was defined by the search space centre (40.50, −6.00, 25.70) and search space size of 25 Å × 25 Å × 25 Å. For cathepsin L, the grid box was defined by the search space centre (55.06, 48.65, 18.12) and search space size of 25 Å × 25 Å × 25 Å.

For both target proteins, the molecular docking of prepared ligands was performed using AutoDock Vina with the exhaustiveness set as 8. The top conformation of the docked ligand with lowest binding energy, i.e., the best docked pose, was obtained from the output of AutoDock Vina using the associated script *vina_split*. Subsequently, a combined structure file in .pdb format of the docked protein-ligand complex (with ligand in the best docked pose) was prepared using a custom script and pdb-tools [79].

### 3.5. Identification of Protein-Ligand Interactions

We searched the combined structure file of the docked protein-ligand complex for ligand binding residues in protein and different non-covalent interactions that can facilitate the binding of the ligand with the protein. These non-covalent protein-ligand interactions were identified using different geometric criteria which are specific to different types of interactions:

Binding site residue. Ligand binding site residues are defined as amino acids in protein which have at least one non-hydrogen atom in the proximity of at least one non-hydrogen atom of the ligand. The distance cut off to determine this proximity between non-hydrogen atoms of protein and ligand is taken to be the sum of their van der Waals radius plus 0.5Å [80].

Hydrogen bonds. The accepted geometric criteria for hydrogen bonds of type D-H⋯A are as follows. Firstly, the distance between the hydrogen (H) and acceptor (A) atom should be less than the sum of their van der Waals radii. Secondly, the angle formed by donor (D), H and A atoms should be >90° (Appendix A). Moreover, carbon (C), nitrogen (N), oxygen (O) or sulfur (S) atoms can be donors while N, O or S atoms can be acceptors [42,81,82].

Chalcogen bonds. In contrast to hydrogen bonds, chalcogen bonds are of type C-Y⋯A, where Y can be a S or selenium (Se) atom and A can be a N, O or S atom. The accepted geometric criteria for chalcogen interactions are as follows. Firstly, the distance between Y and A should be less than the sum of their van der Waals radii. Secondly, the angle formed by the triad, that is ∠C-Y⋯A, should lie in the range 150° to 180° (Appendix A) [83].

Halogen bonds. Halogen bonds are of type C-Y⋯A-B, where halogen Y can be a Fluorine (F), Chlorine (Cl), Bromine (Br) or Iodine (I) atom and A can be a N, O or S atom. The formation of the halogen bond is favoured when the distance between Y and A is ≤3.7 Å and the angle θ_1_ of the A atom relative to the C-Y bond, and the angle θ_2_ of the halogen Y relative to the A-B bond should be ≥90° (Appendix A) [84].

π-π stacking. This interaction occurs between two aromatic rings and can be majorly classified into two types, namely, face-to-face and face-to-edge. In the case of face-to-face type of π-π interaction, the distance between the centroids of the two participating aromatic rings should be <4.4 Å and the angle between their ring planes should be <30°. In the case of face-to-edge type of π-π interaction, the distance between the centroids of the two participating aromatic rings should be <5.5 Å and the angle formed by the ring planes should be in the range 60° to 120° (Appendix A).

Hydrophobic interactions. The geometric criteria for the formation of hydrophobic interactions between atoms in protein and ligand are as follows [85]. The distance between a carbon atom in protein or ligand and a carbon, halogen or sulfur atom in ligand or protein, respectively, should be ≤4 Å. Furthermore, we ensure that the involved atoms in a hydrophobic interaction between protein and ligand do not form hydrogen, chalcogen or halogen bonds between them [85].

In order to detect the above-mentioned protein-ligand interactions, an in-house Python program was written to enable batch processing of combined structure files containing docked protein-ligand complexes for our large phytochemical library.

### 3.6. Comparison with Reference Inhibitors of TMPRSS2 and Cathepsin L

In order to identify potent phytochemical inhibitors of target proteins, we decided to compare the binding energy of the best docked pose of ligands with binding energies of the best docked pose of known inhibitors of TMPRSS2 and cathepsin L obtained from AutoDock Vina.

Recent experiments have shown that both camostat mesylate and nafamostat mesylate, which are approved for human use in Japan, can block the TMPRSS2-dependent cell entry of SARS-CoV-2 [10,15]. By docking these two inhibitors to TMPRSS2 using AutoDock Vina with exhaustiveness set at 50, the predicted binding energies of camostat and nafamostat was found to be −7.4 kcal/mol and −8.5 kcal/mol, respectively. Appendix A show the best docked poses of nafamostat and camostat with TMPRSS2, and it is seen that both molecules form hydrogen bonds with the substrate binding residue D435. Importantly, in comparison to camostat mesylate, nafamostat mesylate in a recent experiment was shown to inhibit the TMPRSS2-dependent cell entry with 15-fold higher efficiency and an EC_50_ value in lower nanomolar range [15], and thus, the docked binding energies of these two known inhibitors are in line with experiments. Based on above observations, we decided on a stringent criteria of docked binding energy ≤−8.5 kcal/mol for screened ligands to be identified as potential inhibitors of TMPRSS2.

Recent experiments have shown that the small molecules E-64d and PC-0626568 (SID26681509) can block the cathepsin L-dependent cell entry of SARS-CoV-2 [10,11,12]. Note that cathepsin L is one of 11 cysteine cathepsin proteases encoded by the human genome, and the cathepsins share a high sequence similarity to papain, a non-specific plant protease [86]. E64-d is a broad spectrum inhibitor which can inhibit proteases cathepsins B, H, L and calpain, while PC-0626568 is a specific inhibitor of cathepsin L [11]. Moreover, a recent study [11] used the specific inhibitor PC-0626568 of cathepsin L to conclude that cathepsin L rather than cathepsin B is important for cell entry of SARS-CoV-2. As both cathepsin L and cathepsin B are expressed in several mammalian tissues, it is important to design specific inhibitors of cathepsin L [75,76] to avoid any off-target toxicity. Along with the above-mentioned two inhibitors of cathepsin L, we have also considered the co-crystallized inhibitor GH4 present in the crystal structure of cathepsin L (PDB 5MQY) as another reference inhibitor of cathepsin L. We remark that while recent experiments [10,11,12] have shown that E-64d and PC-0626568 can block the cathepsin L-dependent cell entry of SARS-CoV-2, similar experimental data specific to SARS-CoV-2 infection is lacking for cathepsin L inhibitor GH4. By docking these—three known inhibitors to cathepsin L using AutoDock Vina with exhaustiveness set at 50, the predicted binding energies of E-64d, PC-0626568 and GH4 were found to be −5.0 kcal/mol, −8.0 kcal/mol and −6.3 kcal/mol, respectively. Notably, the docked binding energies are in line with known specificity of E-64d and PC-0626568 to cathepsin L. Appendix A show the best docked poses of PC-0626568, E-64d and GH4 with cathepsin L. It is seen that both E-64d and PC-0626568 form hydrogen bonds with both catalytic residues C25 and H163. Based on above observations, we decided on a stringent criteria of docked binding energy ≤−8.0 kcal/mol for screened ligands to be identified as potential inhibitors of cathepsin L. We have also compared the docked pose of GH4 obtained from AutoDock Vina with the pose of GH4 in the co-crystallized structure of cathepsin L, and the RMSD between the heavy atoms in the two poses was found to be 0.786 Å. Appendix A shows the superimposed structures of the docked pose of GH4 from AutoDock Vina and the pose of GH4 in the co-crystallized structure of cathepsin L. Apart from PC-0626568 and E-64d, seven other cathepsin-L inhibitors namely, dec-RVKR-CMK, K11777, small molecule 5705213, MDL28170, SSAA09E1, EST and oxocarbazate, have been reported in literature to exhibit ant-coronavirus activity [87].

### 3.7. Molecular Dynamics Simulations

Using GROMACS 5.1.5 [88] along with GROMOS96 54a7 force field [89], we have performed MD simulations for the top inhibitors of TMPRSS2 and cathepsin L to assess the stability of their protein-ligand complexes. The Automated Topology Builder (ATB) version 3.0 (https://atb.uq.edu.au/) was used to generate topology for the top inhibitors [90]. The protein-ligand complex was placed in the center of a cubic periodic box and was solvated by the addition of simple point charge (SPC) waters. Then net charge of the system was neutralized by addition of Na^+^ and Cl^−^ ions. Energy minimization was performed using the steepest descent algorithm. Then the system was heated to 310 K during a constant number of particles, volume, and temperature (NVT) simulation of 500 ps with 2 fs time step. Then the pressure was equilibrated to 1 bar during a constant number of particles, pressure, and temperature (NPT) simulation of 500 ps with 2 fs time step. In both the above simulations, protein and ligand were position restrained. Thereafter, the position restraint was removed and a production MD run was performed for a period of 180 ns with 2 fs time step. During the MD simulation, the structural coordinates were saved after every 2 ps. Temperature was maintained at 310 K using the v-rescale temperature [91]. Pressure was maintained at 1 bar using Parrinello-Rahman pressure coupling method [92]. Time constant for the temperature and pressure coupling were kept at 0.1 ps and 2 ps, respectively. The short-range interactions were computed for the atom pairs within the cutoff of 1.4 nm, whereas the long-range electrostatic interactions were calculated using Particle mesh Ewald (PME) method with fourth order cubic interpolation and 0.16 nm grid spacing. All bonds were constrained using the LINCS method. The trajectories obtained from the MD simulations were then used to compute and analyze the radius of gyration of the protein (R_g_), root mean square deviation (RMSD) of the protein backbone Cα atoms and root mean square fluctuation (RMSF) of the protein backbone Cα atoms using GROMACS scripts.

### 3.8. MM-PBSA Calculation

Molecular Mechanics Poisson-Boltzmann Surface Area (MM-PBSA) method was used to compute the binding energy of the interactions between the protein-ligand complexes for the top inhibitors of TMPRSS2 and cathepsin L. From the 180 ns trajectory obtained from the MD simulation, 80 snapshots were obtained between 100 ns to 180 ns of the simulation at an interval of 1 ns along the trajectory, and thereafter, the binding energy was computed using g_mmpbsa [59,60]. g_mmpbsa computes the Gibbs free energy of binding (ΔG_binding_) using MM-PBSA method as described by the following equation:ΔG_binding_ = G_complex_ − (G_protein_ + G_ligand_)
where G_complex,_ G_protein_ and G_ligand_ represent the total binding energy of the protein-ligand complex, the free protein and the unbounded ligand, respectively.

### 3.9. Validation of the TMPRSS2 Model Structure

In order to validate the model structure of TMPRSS2 built using SWISS-MODEL, we have used two approaches. Firstly, we have run a 180 ns MD simulation of the TMPRSS2 free protein and have computed radius of gyration (R_g_), RMSD and RMSF for the TMPRSS2 free protein from its MD trajectory (Appendix A). The R_g_ of the TMPRSS2 free protein remains largely stable throughout the 180 ns MD simulation (Appendix A). This signifies that the TMPRSS2 model structure remains structurally intact and stable during the MD simulation. The RMSD of the TMPRSS2 free protein becomes stable after 60 ns (Appendix A).

Secondly, we have used the MD trajectory of TMPRSS2 in complex with the top three potential inhibitors (T1, T2 or T3) from 40 ns to 80 ns to validate the TMPRSS2 model structure. The MD trajectory of TMPRSS2 in complex with the inhibitors from 40 ns to 80 ns was used for validation, 80 ns to 100 ns was used for equilibration, and from 100 ns to 180 ns for finding the binding energy of T1, T2 or T3 with TMPRSS2 using MM-PBSA method. To validate the TMPRSS2 model structure, the binding energy computed using MM-PBSA method of T1, T2 or T3 with TMPRSS2 during 40 ns to 80 ns was compared with binding energy of T1, T2 or T3 with TMPRSS2 during 100 ns to 180 ns of the MD simulation. We find that T1, T2 and T3 have average binding energies of −37.507 ± 2.537 kcal/mol, −30.383 ± 4.005 kcal/mol and −26.586 ± 2.506 kcal/mol, respectively, with TMPRSS2 during 40 ns to 80 ns of the MD simulation. The above reported binding energies of T1, T2 and T3 with TMPRSS2 during 40 ns to 80 ns are very close to the binding energies computed for the same inhibitors in complex with TMPRSS2 during 100 ns to 180 ns of the MD simulation as reported in Table 3. These results further validate the TMPRSS2 model structure used for binding energy computations during 100 ns to 180 ns of the MD simulations.

### 3.10. Prediction of ADMET Properties

In order to assess the pharmacokinetic properties and potential toxicity of the inhibitors of TMPRSS2 and cathepsin L predicted from this study, we have also computed the Absorption, Distribution, Metabolism, Excretion and Toxicity (ADMET) properties of the inhibitors using SwissADME [93] and vNN-ADMET [94].

## 4. Conclusions

The current COVID-19 pandemic is a serious threat to humankind. As of 27 July 2020, COVID-19 has led to more than 650,000 deaths worldwide. Due to the absence of approved therapeutics or vaccines against SARS-CoV-2, several countries have been forced to implement partial or complete lockdown measures to restrict infection spread; however, such measures have in turn resulted in an economic catastrophe. Consequently, there is an urgent need to develop antivirals and vaccines against SARS-CoV-2 to protect humankind. In this direction, protein-ligand docking and MD simulation are powerful computational methods to expedite the search for anti-COVID drugs by rapid identification of promising lead molecules. Here, we have used molecular docking and MD simulations in the search of natural compound inhibitors of two human proteases, TMPRSS2 and cathepsin L, that are key host factors in SARS-CoV-2 infection [10,11,12].

Since early civilization, humans have used medicinal plants in different systems of traditional medicine to treat various ailments [95]. Specifically, traditional systems of Indian medicine including Ayurveda, Siddha and Unani have over centuries acquired invaluable knowledge on medicinal plants spanning the rich biodiversity of the subcontinent for treating various ailments including viral infections [28]. As plant-based natural products have been an indomitable source of lead molecules in the course of modern drug discovery [25], it is worthwhile to search for potential anti-COVID drugs among phytochemicals of Indian medicinal plants. In this direction, some of us have built IMPPAT [28], the largest resource on phytochemicals of Indian medicinal plants to facilitate natural product-based drug discovery. In this work, we have performed virtual screening of 14,011 phytochemicals that are produced by Indian medicinal plants to identify potential inhibitors of key host factors, TMPRSS2 and cathepsin L, for SARS-CoV-2 infection. For the identified top inhibitors of TMPRSS2 and cathepsin L, we have performed MD simulation, and thereafter, employed MM-PBSA method to compute binding energies of the protein-ligand complexes.

We have predicted 96 potential phytochemical inhibitors of TMPRSS2, of which the top three candidates, namely, qingdainone (T1), edgeworoside C (T2) and adlumidine (T3), have binding energy value of −39.15 ± 2.799 kcal/mol, −30.284 ± 3.585 kcal/mol and −27.386 ± 2.077 kcal/mol, respectively (Table 3). We have also predicted nine potential phytochemical inhibitors of cathepsin L, of which the top three candidates, namely, ararobinol (C1), (+)-oxoturkiyenine (C2) and 3α,17α-cinchophylline (C3), have binding energy value of −22.384 ± 3.420 kcal/mol, −20.577 ± 3.600 kcal/mol and −26.156 ± 3.433 kcal/mol, respectively (Table 3). Furthermore, most of the herbal sources of the identified phytochemical inhibitors of TMPRSS2 and cathepsin L have been reported to have antiviral or anti-inflammatory use in traditional medicine (Table 1 and Table 2). Additional in vitro and in vivo testing of the identified phytochemical inhibitors of TMPRSS2 and cathepsin L is needed before these molecules can enter clinical trials against COVID-19. In conclusion, we expect the natural product inhibitors identified in this computational study for TMPRSS2 and cathepsin L will likely inform future research toward natural product-based anti-COVID therapeutics.

## Figures and Tables

**Figure 1 molecules-25-03822-f001:**
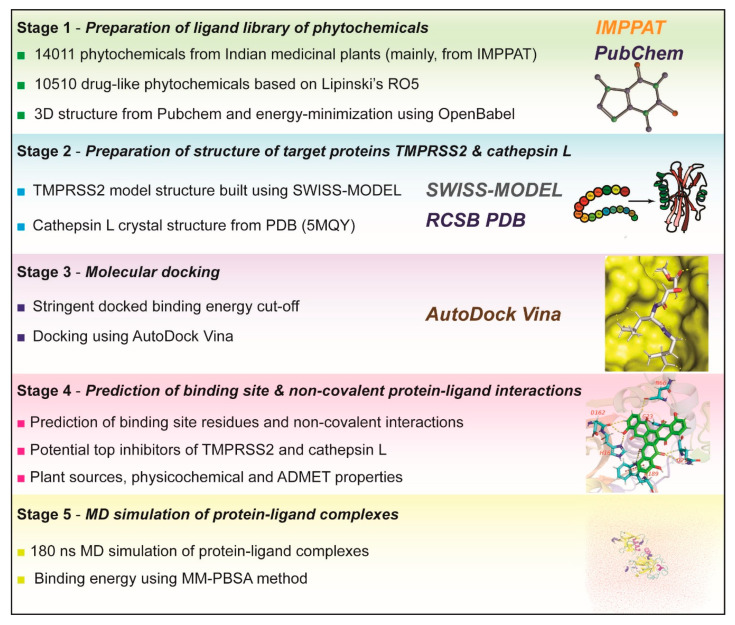
Virtual screening workflow to identify potential phytochemical inhibitors of human proteases TMPRSS2 and cathepsin L.

**Figure 2 molecules-25-03822-f002:**
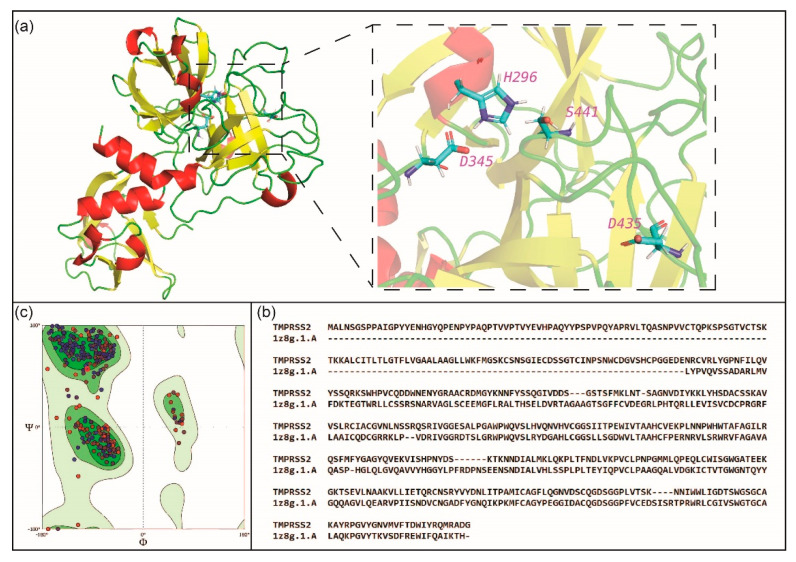
(**a**) Cartoon representation of the homology model structure of TMPRSS2 which has been energy-minimized using UCSF Chimera. The figure zooms into the region containing the catalytic triad Ser441 (S441), His296 (H296) and Asp345 (D345), and the substrate binding residue Asp435 (D435) in the S1 subsite of the enzyme. (**b**) Alignment of protein sequences for TMPRSS2 and hepsin (PDB 1Z8G) which was used as a template to model the structure of TMPRSS2. (**c**) General Ramachandran plot of the energy-minimized model structure of TMPRSS2, which displays the torsional angles, phi (φ) and psi (ψ), of the amino acid residues in the protein.

**Figure 3 molecules-25-03822-f003:**
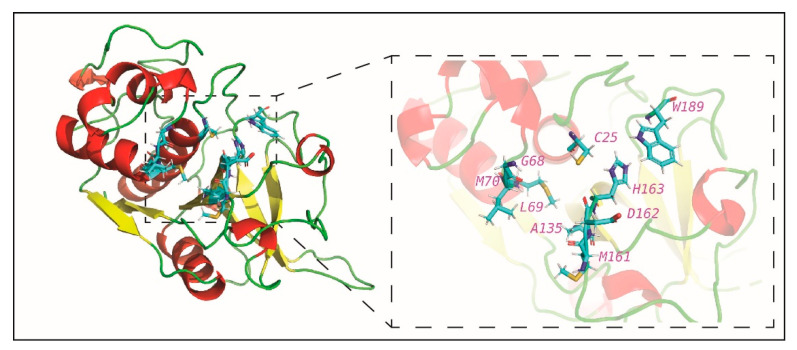
Cartoon representation of the crystal structure of human cathepsin L (PDB 5MQY). The figure zooms into the region containing the catalytic residues Cys (C25) and His163 (H163) in the S1 subsite, residues Asp162 (D162), Met161 (M161), Ala135 (A135), Met70 (M70) and Leu (L69) in the S2 subsite, Trp189 (W189) at the centre of S1′ subsite and the conserved residue Gly68 (G68) in the S3 subsite of the enzyme.

**Figure 4 molecules-25-03822-f004:**
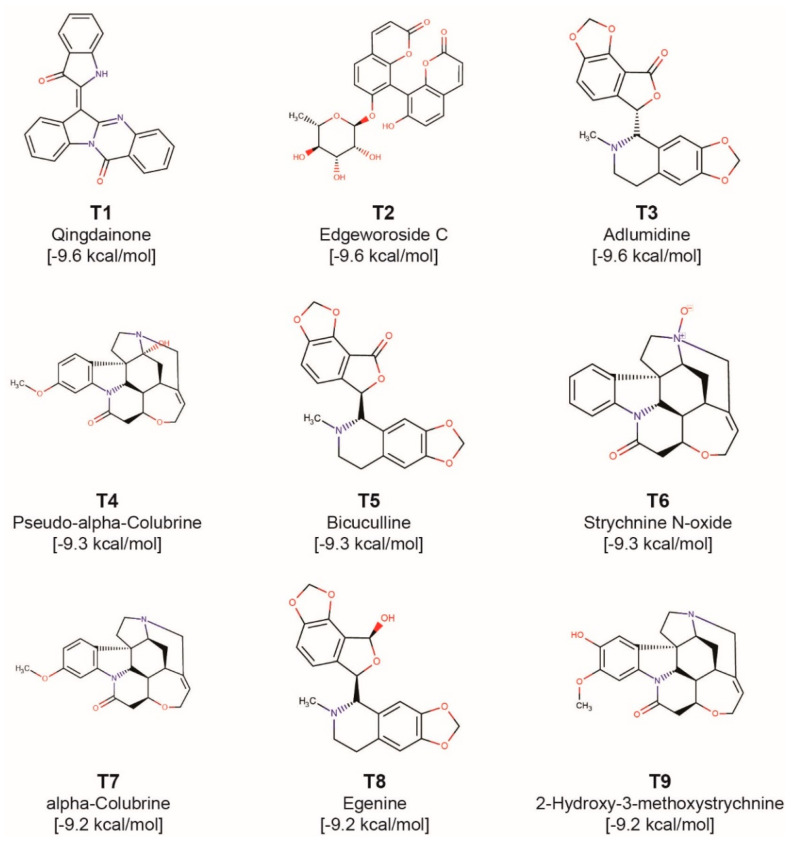
Molecular structures of the top 9 phytochemical inhibitors (compounds **T1**–**T9**) of TMPRSS2. For each inhibitor, the figure shows the 2D structure, common name and docked binding energy of the ligand with TMPRSS2.

**Figure 5 molecules-25-03822-f005:**
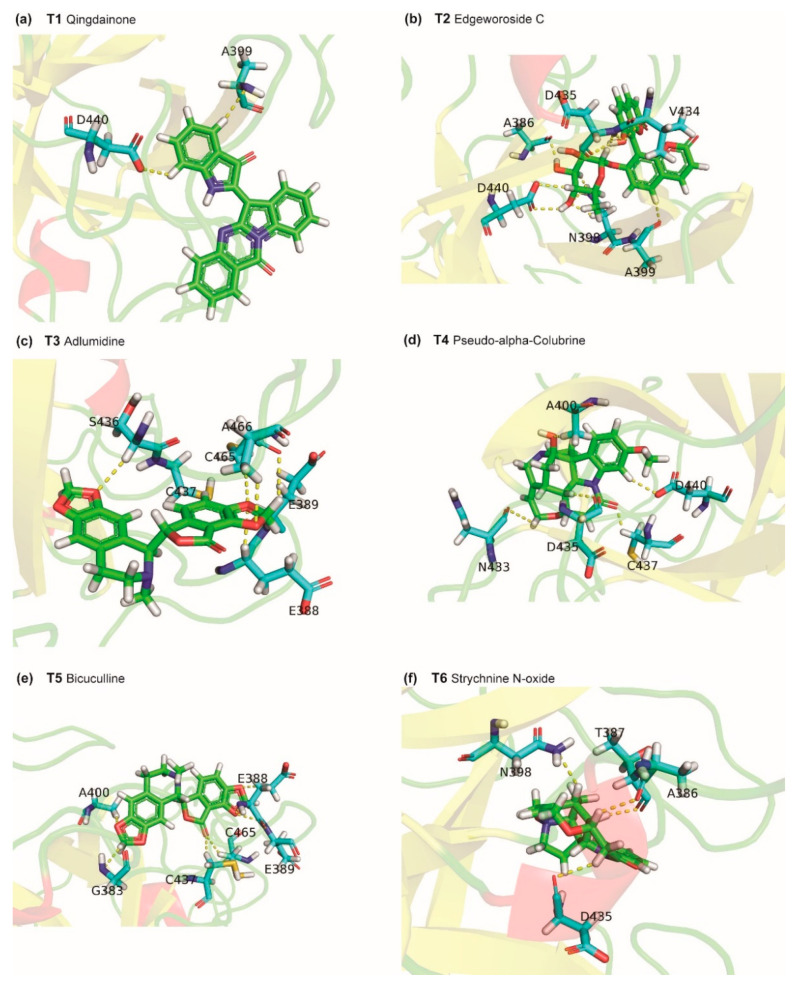
Cartoon representation of the protein-ligand interactions of the phytochemical inhibitors of TMPRSS2. Interactions of TMPRSS2 residues with atoms of (**a**) T1, (**b**) T2, (**c**) T3, (**d**) T4, (**e**) T5, and (**f**) T6. The carbon atoms of the ligand are shown in green colour while the carbon atoms of the amino acid residues in TMPRSS2 are shown in cyan colour. TMPRSS2 residues interacting with the ligand atoms via hydrogen bonds or π-π stacking are labelled with the corresponding single letter residue code along with their position in the protein sequence. The hydrogen bonds and π-π stacking are displayed using yellow and red dotted lines, respectively.

**Figure 6 molecules-25-03822-f006:**
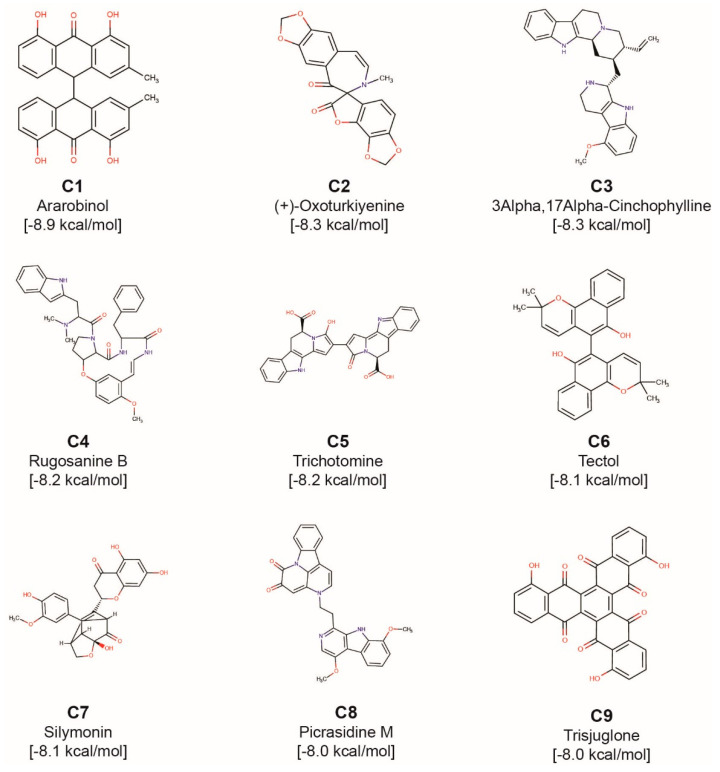
Molecular structures of the top 9 phytochemical inhibitors (compounds **C1**–**C9**) of cathepsin L. For each inhibitor, the figure shows the 2D structure, common name and docked binding energy of the ligand with cathepsin L.

**Figure 7 molecules-25-03822-f007:**
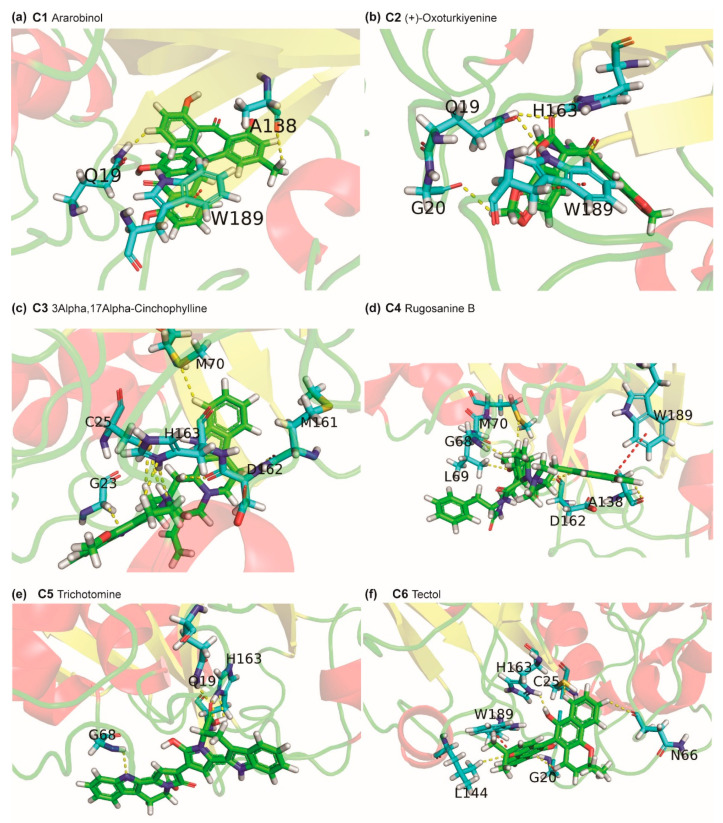
Cartoon representation of the protein-ligand interactions of the phytochemical inhibitors of cathepsin L. Interactions of cathepsin L residues with atoms of (**a**) C1, (**b**) C2, (**c**) C3, (**d**) C4, (**e**) C5, and (**f**) C6. The carbon atoms of the ligand are shown in green colour while the carbon atoms of the amino acid residues in cathepsin L are shown in cyan colour. Cathepsin L residues interacting with the ligand atoms via hydrogen bonds or π-π stacking are labelled with the corresponding single letter residue code along with their position in the protein sequence. The hydrogen bonds and π-π stacking are displayed using yellow and red dotted lines, respectively.

**Figure 8 molecules-25-03822-f008:**
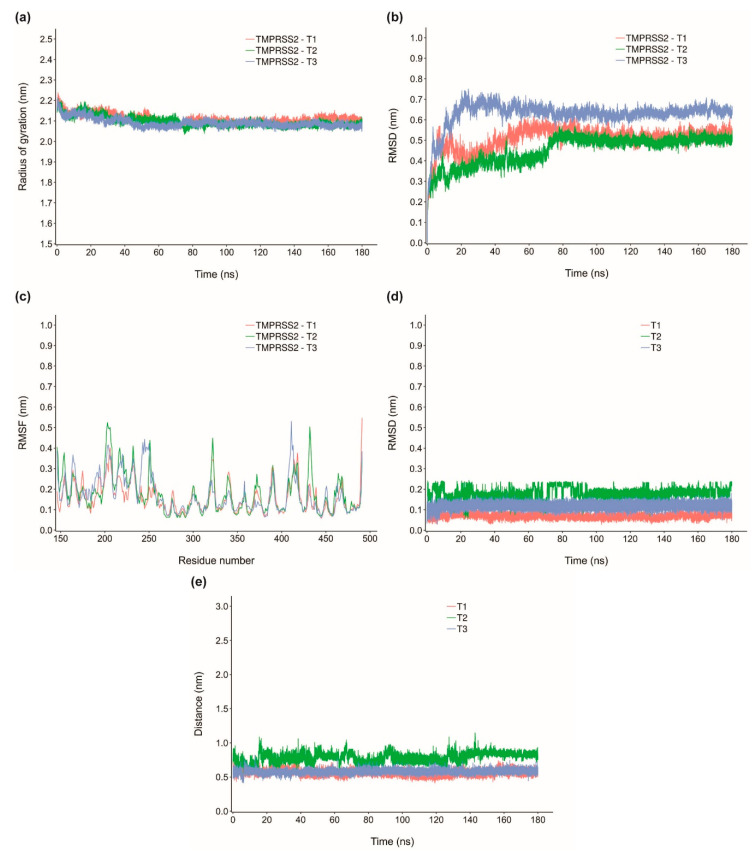
(**a**) Radius of gyration for TMPRSS2 in complex with T1, T2 and T3, (**b**) RMSD for TMPRSS2 in complex with T1, T2 and T3, (**c**) RMSF for TMPRSS2 in complex with T1, T2 and T3, (**d**) RMSD of T1, T2 and T3, and (**e**) Distance of the center of mass of T1, T2 and T3 from the substrate binding residue D435 in TMPRSS2.

**Figure 9 molecules-25-03822-f009:**
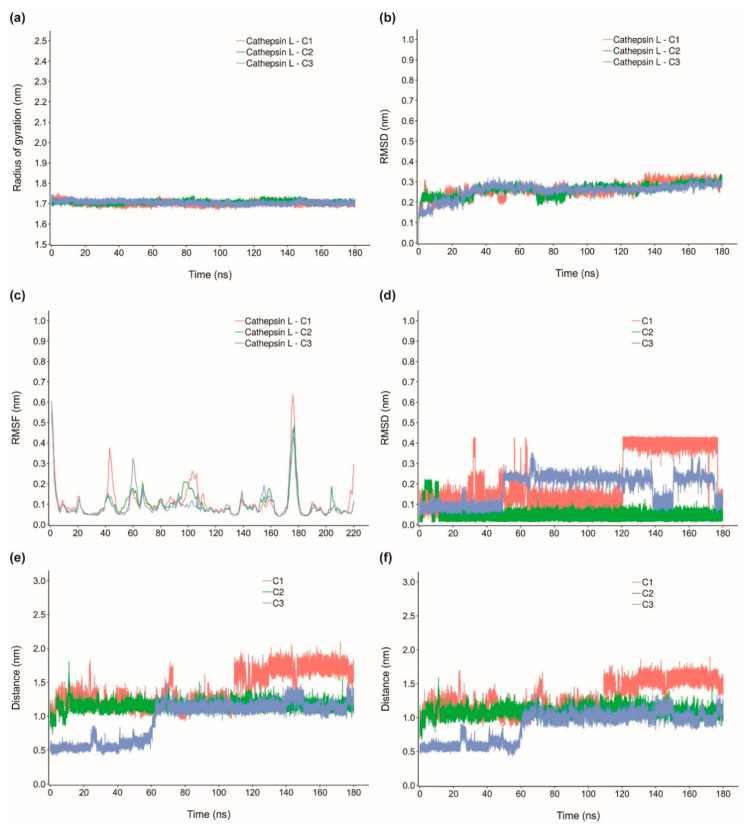
(**a**) Radius of gyration for cathepsin L in complex with C1, C2 and C3, (**b**) RMSD for cathepsin L in complex with C1, C2 and C3, (**c**) RMSF for cathepsin L in complex with C1, C2 and C3, (**d**) RMSD of C1, C2 and C3, (**e**) Distance of the center of mass of C1, C2 and C3 from the catalytic residue C25 in cathepsin L, and (**f**) Distance of the center of mass of C1, C2 and C3 from the catalytic residue H163 in cathepsin L.

**Table 1 molecules-25-03822-t001:** Herbal sources of top 9 phytochemical inhibitors of TMPRSS2. For each phytochemical, the table gives the symbol, docked binding energy, common name and plant source. Plant sources which have been reported to have antiviral or anti-inflammatory use in traditional medicine literature are shown in bold and marked with an [*] sign.

Phytochemical Symbol	Binding Energy (kcal/mol)	Common Name	Plant Source
T1	−9.6	Qingdainone	***Strobilanthes cusia*** **[*]**
T2	−9.6	Edgeworoside C	*Edgeworthia gardneri*
T3	−9.6	Adlumidine	***Fumaria indica*** **[*]**
T4	−9.3	Pseudo-α-Colubrine	***Strychnos nux-vomica*** **[*]**
T5	−9.3	Bicuculline	***Fumaria indica*** **[*]*, Corydalis govaniana* [*]*, Nerium oleander* [*]**
T6	−9.3	Strychnine *N*-oxide	***Strychnos nux-vomica*** **[*]*, Strychnos ignatii* [*]*, Strychnos colubrina* [*]**
T7	−9.2	α-Colubrine	***Strychnos nux-vomica*** **[*]*, Strychnos ignatii* [*]*, Strychnos colubrina* [*]**
T8	−9.2	Egenine	***Fumaria vaillantii*** **[*]**
T9	−9.2	2-Hydroxy-3-methoxystrychnine	***Strychnos nux-vomica*** **[*]**

**Table 2 molecules-25-03822-t002:** Herbal sources of top 9 phytochemical inhibitors of Cathepsin L. For each phytochemical, the table gives the symbol, docked binding energy, common name and plant source. Plant sources which have been reported to have antiviral or anti-inflammatory use in traditional medicine literature are shown in bold and marked with an [*] sign.

Phytochemical Symbol	Binding Energy (kcal/mol)	Common Name	Plant Source
C1	−8.9	Ararobinol	***Senna occidentalis*** **[*]**
C2	−8.3	(+)-Oxoturkiyenine	*Hypecoum pendulum*
C3	−8.3	3Alpha,17Alpha-Cinchophylline	***Cinchona calisaya*** **[*]**
C4	−8.2	Rugosanine B	***Ziziphus rugosa*** **[*]**
C5	−8.2	Trichotomine	***Clerodendrum trichotomum*** **[*]**
C6	−8.1	Tectol	***Tectona grandis*** **[*]*, Tecomella undulata* [*]**
C7	−8.1	Silymonin	***Silybum marianum*** **[*]**
C8	−8	Picrasidine M	***Picrasma quassioides*** **[*]**
C9	−8	Trisjuglone	***Juglans regia*** **[*]**

**Table 3 molecules-25-03822-t003:** MM-PBSA based binding energy for top three inhibitors of TMPRSS2 and cathepsin L.

Protein-Ligand Complex	Binding Energy (kcal/mol)	Van Der Waals Energy (kcal/mol)	Electrostatic Energy (kcal/mol)	Polar Solvation Energy (kcal/mol)	SASA Energy (kcal/mol)
TMPRSS2-T1	−39.15 ± 2.799	−54.285 ± 2.903	−3.031 ±1.439	22.844 ± 2.44	−4.678 ± 0.237
TMPRSS2-T2	−30.284 ±3.585	−49.048 ± 3.838	−12.501 ± 4.884	35.978 ± 5.226	−4.712 ± 0.320
TMPRSS2-T3	−27.386 ± 2.077	−39.379 ± 2.109	−8.846 ± 1.423	24.359 ± 2.157	−3.52 ± 0.210
cathepsin L-C1	−22.384 ± 3.420	−25.296 ± 3.127	−2.214 ± 1.661	7.988 ± 4.103	−2.861 ± 0.366
cathepsin L-C2	−20.577 ± 3.600	−30.129 ± 3.154	−4.572 ± 2.138	16.891 ± 3.533	−2.767 ± 0.234
cathepsin L-C3	−26.156 ± 3.433	−37.165 ± 3.308	−2.093 ± 1.379	16.958 ± 4.513	−3.856 ± 0.319

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
