# Peer review of "In Silico Identification of Potential Natural Product Inhibitors of Human Proteases Key to SARS-CoV-2 Infection"

_molecules, 2020, doi:10.3390/molecules25173822_

Round 1

Reviewer 1 Report

The authors of the manuscript titled "In silico identification of potential natural product inhibitors of human 3 proteases key to SARS-CoV-2 infection" have investigated the potential usefulness of Indian natural products for the discovery of inhibitors of two host proteases critical for the entry of the SARS-CoV2 virus into host cells. They have employed a combination of computational approaches in their study, including database curation for the natural product ligands and for the coordinates of human Cathepsin L, homology modeling to obtain 3D coordinates for the human TMPRSS2 protein, ligand docking using Autodock Vina, molecular dynamics simulation using Gromacs and the Gromos53a7 forcefield, MMPBSA calculation for MD-generated snapshots of selected protease-'inhibitor' complexes, as well as calculation of druggability scores. In the overall, the work was well designed, well written, well presented, and make for an interesting read. Regardless of these, there are a number of points and issues that I will like to point the authors' attention to:

1.) Correction of few grammatical errors. E.g. "Preparation" instead of "Prepararion" in the flowchart in Figure 1.

2.) The flowchart is too wordy and thus cumbersome to read. Flowcharts are better appreciated when they communicate textual information graphically with as few words as possible. The authors are encouraged to amend wherever possible.

3.) Low sequence similarity in the homology modeling part. A 38% sequence similarity reported for the SWISSModel-modeled human TMPRSS2 protein is rather low for structure-based inhibitor design work where values above 50 % are generally preferred (See Drug Discov Today. 2004 Aug; 9(15): 659–669.) The least the authors could do to properly validate such kind of prediction is to conduct extensive explicit solvent MD simulation on the model and then analyse structural descriptors of stability to obtain some decent assessment of the quality of the model. If the authors have done this, then they need to create a section in the manuscript that directly addresses this.

4.) Irregular scale for plots in Figure 8 (a) and (b): The authors should use a common scale for the y-axis of the specified figures to allow for easy comparison.

5.) MD observables: authors should present the distance-dependent RMSD values for the bound compounds alone with least square fitting to the docked pose. What the authors have presented in the current rendition is the RMSD of ligand-protein complexes which failed to show clearly if the binding predictions are indeed temporally stable.

6.) MD observables: authors should present the distance-dependent distance (e.g. center of mass) of the ligands from the catalytic sites of the two proteases.

7.) The authors have failed to present the docking result for cathepsin L and the co-crystallized inhibitor present in the crystallographic structure used.

8.) The authors in discussing their docking results cited a number of the compounds that had been reported with antiviral activities, such as Ararobinol with existing patent for the human influenza virus. Authors should justify how such vague allusions should be taken as validation for the data presented here by them. For instance, stating that a compound has been reported with antiviral activities fails to indicate the relationship between the reported antiviral activities with the proposed inhibition of TMPRSS2 and Cathepsin L being reported by the authors. Specific viruses and targeted enzymes underlying the previous antiviral claims should be included by the authors.

Reviewer 2 Report

The manuscript entitled ‘In silico identification of potential natural product inhibitors of human proteases key to SARS-CoV-2 infection’ is quite interesting, but authors should make some changes to improve the paper's quality:

1.Please, provide more information about known TMPRSS2 and cathepsin L inhibitors

2. Please, provide more information about putative molecular targets of identified candidates. It is worth mentioning for example that adlumidine is a ligand of GABA receptor (J. Med. Chem. 1992, 35, 11, 1969–1977)

Round 2

Reviewer 2 Report

I am generally satisfied with the revisions, and I recommend the acceptance of this work.